# PAX7, a Key for Myogenesis Modulation in Muscular Dystrophies through Multiple Signaling Pathways: A Systematic Review

**DOI:** 10.3390/ijms241713051

**Published:** 2023-08-22

**Authors:** Nor Idayu A. Rahman, Chung Liang Lam, Nadiah Sulaiman, Nur Atiqah Haizum Abdullah, Fazlina Nordin, Shahrul Hisham Zainal Ariffin, Muhammad Dain Yazid

**Affiliations:** 1Centre for Tissue Engineering & Regenerative Medicine, Faculty of Medicine, Universiti Kebangsaan Malaysia Medical Centre, Jalan Yaacob Latif, Cheras, Kuala Lumpur 56000, Malaysia; idayurahman@ukm.edu.my (N.I.A.R.);; 2Centre of Biotechnology & Functional Food, Faculty of Science and Technology, Universiti Kebangsaan Malaysia, Bangi 43600, Malaysia

**Keywords:** signaling pathways, muscular dystrophy, *Pax7*, myogenesis

## Abstract

Muscular dystrophy is a heterogenous group of hereditary muscle disorders caused by mutations in the genes responsible for muscle development, and is generally defined by a disastrous progression of muscle wasting and massive loss in muscle regeneration. *Pax7* is closely associated with myogenesis, which is governed by various signaling pathways throughout a lifetime and is frequently used as an indicator in muscle research. In this review, an extensive literature search adhering to the Preferred Reporting Items for Systematic Reviews and Meta-Analyses (PRISMA) guidelines was performed to identify research that examined signaling pathways in living models, while quantifying *Pax7* expression in myogenesis. A total of 247 articles were retrieved from the Web of Science (WoS), PubMed and Scopus databases and were thoroughly examined and evaluated, resulting in 19 articles which met the inclusion criteria. Admittedly, we were only able to discuss the quantification of *Pax7* carried out in research affecting various type of genes and signaling pathways, rather than the expression of *Pax7* itself, due to the massive differences in approach, factor molecules and signaling pathways analyzed across the research. However, we highlighted the thorough evidence for the alteration of the muscle stem cell precursor *Pax7* in multiple signaling pathways described in different living models, with an emphasis on the novel approach that could be taken in manipulating *Pax7* expression itself in dystrophic muscle, towards the discovery of an effective treatment for muscular dystrophy. Therefore, we believe that this could be applied to the potential gap in muscle research that could be filled by tuning the well-established marker expression to improve dystrophic muscle.

## 1. Introduction

Muscular dystrophy is a heterogeneous group of inherited muscle disorders affecting both sexes and races worldwide, with varying prevalence for each type [1]. Muscular dystrophy is largely attributed to mutations occurring in the genes responsible for muscle development, which are mostly inherited either through X-linked, autosomal dominant, or autosomal recessive inheritance [2,3,4]. In small percentages, muscular dystrophies are caused by de novo mutations [5]. The underlying pathogenetic factors greatly vary, translating to the variability in the affected protein in different types of muscular dystrophy. Loss of these genes compromises structural integrity of the muscle fibers. This results in the weakening of muscle over time and burdens patients by gradually decreasing their quality of life. There are more than thirty types of muscular dystrophies, with nine most prevalent forms, distinguishable by age of onset, progression rate, and symptoms that influence the prognosis of disease severity and lifespan expectancy [6,7]. Muscular dystrophy can be characterized, although this varies, by the progression of muscle wasting and loss of muscle regeneration capabilities, leading to fragile fibers due to impaired myogenesis [8]. In the case of Duchenne Muscular Dystrophy, the catastrophic progression of muscle wasting leads to premature death [9].

Myogenesis is a series of progressive development of skeletal muscle tissue over a lifetime where myoblasts, the early mononucleated committed precursor cells of skeletal muscle fuse together and differentiate into myotubes, the multinucleated muscle cells that later undergo further differentiation and fusion to form myofibers [10]. Myoblast heterogeneity stems from three types of myoblasts: embryonic, fetal, and satellite cells with distinct genetic backgrounds that are traditionally distinguished by desmin, myogenin (MyoG), and myosin heavy chain isoform (MyHC) expression [11,12]. These myoblast transitions are thought to overlap at multiple points during myogenesis, due to the activation of several factors. Four myogenic regulatory factors (MRF), Myogenic factor 5 (Myf5), Mrf4, Myogenic Differentiation 1 (MyoD), and MyoG play critical roles in the precise differentiation of progenitor myoblasts into myofibers during embryonic-to-adult myogenesis [13,14].

*Pax7* is a transcription factor known to be indispensable in myogenesis by modulating *MyoD* expression and maintaining healthy and mature cell differentiation [15,16]. *Pax7* belongs to the highly conserved paired-box (Pax) gene family transcription factor and is critically expressed during the development of the nervous and muscular systems [17]. Interestingly, it has been revealed that overexpression of *Pax7* regulates myogenic differentiation by downregulating *MyoD* expression and prevents *MyoG* induction in fetal Duchenne Muscular Dystrophy (DMD) muscle tissues, suggesting a delay in myofibers differentiation program [18,19].

Several major signaling pathways are intricately linked to each other in governing diseases, making it challenging to understand the orchestration of cellular events upon changes in gene expression. Therefore, in this review, we highlight the thorough evidence of Pax7′s precise function in multiple signaling pathways. Moreover, this systematic review will provide insight into a decade of the relevant research evidence in Pax7 involvement in modulating myogenesis and deciding cell fate, with additional stress on the novel approach that could be taken in manipulating Pax7 expression itself in dystrophic muscle towards the discovery of effective treatment for muscular dystrophies.

## 2. Methods

### 2.1. Search Strategy

This systematic review is performed according to the Preferred Reporting Items for Systematic Reviews and Meta-Analyses (PRISMA) 2020 protocol and guidelines [20]. We accessed and evaluated the articles with respect to *Pax7* gene expression function in modulating myogenesis in in vitro and in vivo studies for their analyses associated with signaling pathways. Three independent searches of electronic databases, the Web of Science (WoS), PubMed and Scopus were utilized on 16 December 2022, using the keywords of “muscular dystrophy” AND “Pax7” AND “myogenesis” to identify the studies for further analysis.

### 2.2. Eligibility Criteria

We further refined the articles to those published within 10 years (2013–2022), with the preliminary screening list filtered down to in vitro and in vivo experiments, regardless of the animal model used. We screened for original articles written in English and excluded review articles, conference abstracts, letters, book chapters, case studies, reports, and editorials, including method and protocol articles. Two investigators independently reviewed the titles and abstracts for eligibility, and the entire texts were reviewed to ascertain eligibility. Articles that observed the signaling pathway and *Pax7* gene and/or protein expression in their research were chosen for our review, irrespective of their methodological approaches. The post-screening selected articles then underwent abstract screening and auditing. Exclusion criteria were applied, in which duplicates and outcomes irrelevant to our objective were excluded.

### 2.3. Study Selection

Keyword searches on three independent websites yielded 247 articles, of which 99 were from the Web of Science (WoS), 81 from PubMed, and 67 from Scopus. The initial screening of research works focusing on muscular dystrophies from 2013 to 2022, as well as the elimination of duplicates resulted in 124 articles: 69 from Web of Science (WoS), 34 from PubMed, and 21 from Scopus. The total number of articles eligible for this review based on inclusion and exclusion criteria is 91. The articles were further screened for inclusion in the study, and the entire text was read to determine eligibility, if necessary. As a result, a total of 19 articles were ultimately included in the review for dissection and extraction. The inclusion process for this review is visually presented in Figure 1, and Appendix A are provided to address the exclusion criteria.

### 2.4. Data Collection

The risk-of-bias assessment was performed (Table 1) and generated by two reviewers independently to reduce individual and selection bias/concern during the article selection process. The risk-of-bias domains were categorized as definitely low risk of bias, probably low risk of bias, probably high risk of bias, definitely high risk of bias and not applicable.

## 3. Results and Discussion

### 3.1. Pax7 as a Key Indicator for Myogenesis

*Pax7*, also known as Paired-Box Gene-7, is a transcription factor that belongs to the PAX family and is strongly linked to adult satellite cells, which are precursors of skeletal muscle cells [16,40]. Satellite cells are derived from embryonic progenitor cells during development, which divide only a limited number of times, whereas adult satellite cell activation induces differentiation, proliferation, and fusion in muscle regeneration, transforming it into myofibers [41,42]. Quantifying the *Pax7* gene and/or protein expression as a marker for determining the function of a novel molecular factor at different phases of myogenesis is unavoidable in muscular dystrophy research. *Pax7*-expressing satellite cells are well-established models for determining myoblast activation and myogenic status [43,44]. *MyoD* belongs to the basic helix-loop-helix transcription factor family, and is frequently used as a marker for myogenic and activated satellite cell identification in conjunction with Pax7 [44,45].

*Pax7* and *MyoD* interact with each other during myogenesis, making them ideal markers for determining the cells’ quiescent, activation and myogenic status. The *Pax7* and *MyoD* proportions of satellite cell subpopulations have well-established symmetric expression patterns; Pax7+/MyoD−, Pax7+/MyoD+, and Pax7−/MyoD+ [44,46,47]. The Pax7+/MyoD− cell population pattern, known as self-renewal satellite cells, highly expresses *Pax7* to keep the cells quiescent, and is necessary for the maintenance of stem cell homeostasis [43,48]. Pax7+/MyoD+ cells are activated satellite cells that express myogenic factors and permit the cells to differentiate and proliferate, whereas Pax7−/MyoD+ cells are differentiating cells that are also controlled by other MRFs [49]. The symmetric expression pattern quantification is also often quantified using *Pax7* and *Myf5* instead of *MyoD*.

To investigate muscle regenerative capacity, Fujimaki et al. and Pisconti et al. measured *Pax7* gene and/or protein expression levels in satellite cells during all phases of myogenesis before and after injury. Differential expression of *Pax7* as a proliferation marker was observed at different degrees of injury to reveal the state of the satellite cell niche, where Fujimaki et al. discovered that Pax7+ cell number per myofiber in satellite cell-specific conditional knockout mice for *Notch1/Notch2* (scDKO) is almost entirely ablated compared to control mice after 5 and 19 days of final tamoxifen (TMX) treatment, suggesting impaired muscle regeneration. Pax7+/MyoD− cell population expansion in scDKO mice was suppressed, which agreed with the reduction of proliferative cells and induced *MyoG* expression, compared to control mice. Fiore et al. investigated the muscle fiber regenerative capacity even after acute injury by observing the exogenous and endogenous expression of *Pax7*, which reflects the self-renewable and myogenic ability of satellite cells at different ages of the mice model. The team quantified the number of Pax7+ cells in tibialis anterior (TA) sections of the mice at different ages, of 6 weeks, 12 weeks, 6 months, and 12 months old, and compared calcium-dependent and phospholipid-dependant protein kinase (PKCθ) null mice with the controls. They found that PKCθ null mice showed over 50% more Pax7+ cells than the controls. The presence of Pax7+/MyoD− and Pax7+/MyoD+ cell populations implied an increase in both quiescent and proliferative cells in PKCθ null mice. Furthermore, Pisconti et al. discovered that the cell population, Pax7−/Myf5+, was approximately 25% higher in transmembrane heparan sulphate proteoglycan syndecan-3 (*Sdc3*) null dystrophic mice (Sdc3−/−) than in wild-type (WT), indicating that reduced *Pax7* expression in the absence of *Sdc3* resulted in a reduction of self-renewable cells. In contrast, the Pax7+/Myf5+ population proportion in Sdc3−/− mice was nearly twofold that of WT, indicating an increase in proliferating myogenic progenitors, and histopathology analysis revealed improved muscle fibrosis followed by decreased sarcolemmal permeability.

Servián-Morilla et al. discovered the D233E mutation on *POGLUT1*, a gene asssociated with enzymatic activity related to glycosylation pathway(s). α-dystroglycan is an essential component of the dystrophin-glycoprotein complex at the sarcolemmal junction which connects the extracellular matrix to the cytoskeleton through *Laminin-2* [50]. Any mutation in *POGLUT1* may induce disruption in α-dystroglycan, which therefore compromises the integrity of the dystrophin-glycoprotein complex, leading to degenerative muscle diseases including DMD [51,52,53]. The mutation resulted in a decrease in *Pax7* mRNA level and Pax7+ cells, indicating a loss of the self-renewal ability of satellite cells in the patients’ muscle compared to controls, presumably due to the flawed O-glucosylation. They thus propose that impaired myogenesis mediated by inactivation of Notch signaling is a causal mechanism of muscular dystrophy in siblings in relation to the post-translational modification and maintenance of muscle stem cells, leading to muscular degeneration and α-dystroglycan hypoglycosylation. The essential role of measuring Pax7 in muscular dystrophy research has been demonstrated in all phases of muscle development, in the effort to characterize some novel molecular factor functions, as it is dependent on the ability of satellite cells to regenerate.

### 3.2. Pax7 in the Context of Embryonic, Fetal, and Adult Myogenesis

Muscle development occurs as a sequential event, starting with embryonic, to fetal, and ending with adult myogenesis. The majority of embryonic myoblasts and satellite cells contain Paired-Box Gene-3 (Pax3), a member of the PAX family of transcription factors, which is a pivotal gene involved in organ development [13,54]. Satellite cells which are capable of self-renewal and myogenic differentiation facilitate postnatal muscle formation and regeneration. The remnants of embryonic-origin *Pax7*-expressing cells from development give rise to post-embryonic myogenesis, and are required for fetal myogenesis as well as adult skeletal muscle regeneration, which is critical for skeletal muscle repair. Using a dy^W^ mice model, which is derived from homologous recombinant in mouse embryonic stem (ES) cells with laminin alpha 2 (*Lama2*)-knockout [55], Nunes et al. explored the *merosin*-deficient congenital muscular dystrophy type 1A (MDC1A) to trace the advent of the disease during muscle development in utero. They observed the homozygous dy^W^ mice embryo to have significantly smaller muscle than the wild type at the fetal age of E15.5 to E18.5, although expressing the same number of myofibers. They discovered that the increase in the Janus kinase/signal transducers and activators of the transcription (JAK-STAT) signaling pathway, together with the decreased expression of *myostatin* and *Pax7*, caused the satellite cells to lose their normal level of self-renewal capacity. Nunes et al. also discovered an increase in *MyoD* expression at E18.5, as the cells exited the cell cycle faster, and reduced the differentiation and fusion rate. dy^W^ mice had an aging phenotype that could not recover during the development event, due to low *Pax7* levels, reflecting a low level of self-renewal capacity. Fetal myogenesis is closely associated with the formation of myofibers (hyperplasia) and myofiber fusion (hypertrophy). Improper execution of this sequential event may result in fewer differentiated and competent cells in utero.

Farini et al. investigated the muscle development stage in mice model fetuses and muscular biopsies from 12-week-old DMD and healthy human fetuses, using histological evaluation. They revealed a decrease in fiber density, where dystrophic fetal muscles expressed significantly higher levels of fast-type genes such as sarcoplasmic reticulum calcium ATPase-1 (*ATP2A1*), troponin T type 3 (*TNNT3*) and troponin fast C Type 2 (*TNNC2*). Dystrophic muscle is known to have dystrophin protein loss, which causes membrane tears and sarcolemmal destabilization, resulting in abnormal Ca^2+^ reflux [56,57]. Farini et al. examined myogenic marker expression in 12-week-old healthy and DMD human fetal muscles, and discovered a significantly higher number of Pax7+ cells per myofiber in DMD samples compared with healthy. *PKCα* is a serine/threonine kinase belonging to the PKC family, and it is involved in numerous cellular processes. In human fetal dystrophic muscle, *PKCα* is upregulated, and its activity is regulated by Ca^2+^ levels, contributing significantly to muscle development and plasticity. They hypothesized that inositol 1,4,5-trisphosphate (IP3)-IP3R binding-mediated Ca^2+^ signaling determines calcium cell accumulation in DMD muscle. Farini et al. discovered that enhanced *PKCα* activation causes altered myogenesis in DMD muscles, by overexpressing *Pax7* while suppressing *MyoD*, along with a delay in fiber maturation and a modification in fiber type composition. Therefore, by attuning the IP3/IP3R pathway, satellite cells could exit the stemness state and express myogenic markers accordingly, during myogenesis, for the next regeneration cycle, and limit DMD muscle damage by Ca^2+^ deposition.

Tierney et al. in their research in 2016 demonstrated the extracellular matrix (ECM) function in muscle stem cell regulation at different stages of muscle development. Myogenesis regulation in prenatal muscle stem cells focuses primarily on cellular contribution to myofibers, laying the foundation for subsequent muscle growth. Conversely, in postnatal or adult muscle stem cell myogenesis, the regulation is more focused on the balance between muscle growth and repair, as well as the maintenance of the stem cell pool population for the next regeneration. Fetal muscle stem cells adapt to the microenvironment by autonomously secreting ECM. This leads to stem cell expansion support and enhances stem cell function intrinsically by actively dividing and maintaining the *Pax7* expression level in growing conditions. They revealed that in differentiation conditions, *Pax7* expression increased while *Myf5* and *MyoG* expression decreased, showing that fetal muscle stem cells resist myogenic lineage progression. This implies that fusion efficiency increases during development. Fetal muscle stem cells expressed higher level of Notch targets and a reduced level of canonical Wnt targets, compared to activated adult muscle stem cells, suggesting innate and robust expansion. Adult muscle stem cells profited from co-transplantation of fetal and adult muscle stem cells by boosting *Pax7* expression. ECM molecule remodeling was influenced by fetal stem cells in a paracrine manner, improving adult stem cells’ regenerative potential for long-term self-renewal, implying that fetal muscle stem cells are responsible for local microenvironmental remodeling and promoting neighboring adult muscle stem cells’ regenerative potential. They concluded that fetal muscle stem cells have enhanced regenerative potential during tissue repair and the ability to repopulate the stem cell pool during development. Fetal muscle stem cells possess remarkable regenerative potential through more efficient expansion and selective expression of ECM proteins to remodel their local microenvironment.

In a context of adult myogenesis, Jiang et al. investigated how satellite cell number and activity decline with age in the mdx mice model. The mdx model is a universal mice model used as DMD murine strain with a point mutation in the *dystrophin* gene, and exhibits DMD phenotypes [58,59,60]. They found that mdx mice expressed higher *Pax7* than in the wild-type, at different age groups. Their findings were suggested severe age-dependent deficiencies in satellite cell activation and/or proliferation, causing the mice to have reduced satellite cell self-renewal capacity, due to ongoing muscle injuries. Mdx mice had a decrease in Pax7+/MyoD− cells but an increase in MyoG+/MyoD+ cells, indicating terminal differentiation. They demonstrated that aberrant Notch signaling, which was supposed to be activated and improve the self-renewal capacity, was actually responsible for defective satellite cell renewal in dystrophic muscles. *Dystrophin* and the Notch signaling cascade may be perturbed in mdx mice satellite cells in dystrophic muscles, as they play an important role in embryonic muscle formation and postnatal myogenesis. The Notch signaling pathway is inhibited in dystrophic muscle satellite cells, leading to a defective self-renewal capacity and potential side effects. The research demonstrated the significance of quantifying *Pax7* in the sequential event of myogenesis from embryonic and fetal, to adult myogenesis, to understand the relevance of certain signaling pathways and the role of a novel molecular factor in the rescue of dystrophic muscle.

### 3.3. The Signaling Pathways Underlying Satellite Cell Regulation

#### 3.3.1. Notch

Notch signaling is a juxtracrine cellular signaling that interacts on the cell surface via ligand–receptor crosstalk. Notch signaling is a multifunctional pathway that is critical in cell fate decisions during muscle development and regeneration. It is also frequently associated with satellite cell proliferation, differentiation, and self-renewal. Notch signaling dysregulation may contribute to various developmental disorders, demonstrating the importance of this signaling in living organisms [61,62]. Fiore et al. and Pisconti et al. explored the absence of a *PKCθ* and *Sdc3* in dystrophic muscles respectively, in in vitro and in vivo models using mice. The absence of *PKCθ* in dystrophic mice increases muscle regeneration, and reduces muscle necrosis and fibrosis, while improving exercise performance without exhaustion of satellite cells, via the activation of Notch signaling and the increase in the Pax7+ quiescent satellite cells. The loss of *Sdc3* caused the decrease in Pax7+ cells and inhibited Notch signaling. Fujimaki et al. used a different approach in investigating the Notch signaling pathway in muscular dystrophy, by modulating the parallel expression of *Notch1* and *Notch2* to maintain the quiescent state of satellite cells after injury. They revealed that *Notch1* and *Notch2* are essential for muscle regeneration after injury, by tightly regulating *Pax7* expression throughout myogenesis. *Notch1* and *Notch2* cooperate and compensate for one another to control muscle regeneration in adults, hence sustaining the satellite cell pool.

Bi et al. investigated the role of Notch signaling in post-fusion myofibers and muscle regeneration using stage-specific activation of Notch1 signaling in mouse strains Myl1^Cre^ and MCK-Cre. They discovered that post-differentiation myocytes can de-differentiate back into Pax7+ quiescent satellite cells via Notch1 signaling activation, which is stage-dependent and restricted to mononucleated myofibers, which do not require cytokinesis. Activating Notch signaling in aged and dystrophic mice enhanced muscle function and regeneration; however, it caused severe muscle damage and postnatal mortality, emphasizing the stage-specific nature of *Notch1* in maintaining the stem cell niche.

A contrasting approach taken by Mu et al. in manipulating Notch signaling expression was shown via the inhibition of overexpressed Notch. The inhibition rescued the induction of myogenic differentiation and improved the function of *dystrophin*/*utrophin* double knockout (dKO) mice muscle progenitor cells (MPC). As a result, Notch overexpression has been shown to negatively regulate muscular dystrophy phenotypes, and re-balancing Notch signaling in dKO mice is critical for modulating pro-inflammatory factors, cell proliferation, and the rate of MPC senescence. Despite the fact that Notch activation facilitates regeneration, Wen et al. 2012 [63] discovered that the ability to increase satellite cells through self-renewal may inhibit muscle regeneration in old mice by delaying myogenic activation. George et al. demonstrated the importance of Numb protein, a Notch signaling inhibitor in muscle regenerative myogenesis expressed in activated satellite cells. Numb deficiency in the Pax7-lineage decreased the number of satellite cells in single-fiber culture, resulting in a defective repair response in adult muscle. Numb-deficient cells show no substantial change in the expression levels of *MyoD*, *Pax7*, and *Notch* ligands, stipulating that Notch signaling is not the major mechanism regulating satellite cell proliferation. Numb regulates skeletal muscle repair, asymmetric satellite cell division, and activation-associated satellite cell progeny proliferative expansion.

Coppens et al. performed a cohort study with 23 participants from 13 unconnected families with bi-allelic *JAG2* variants, and found that the majority of participants had a distinct pattern of progressive muscle weakness in the lower extremity and proximal dominance muscle. *JAG2* is a direct target in the Notch signaling pathway, and is frequently associated with muscular dystrophy [64,65]. Notch signaling is essential for maintaining satellite cell quiescence in adult muscle, with *JAG2* and *Notch3* being the most abundant. Despite *Jagged1* and *Serrate*, other ligands in Notch signaling which have therapeutic potential for muscular dystrophy, *Jag2* knockout mice models died prematurely, due to a cleft palate. *JAG2*-related muscular dystrophy is linked with bi-allelic pathogenic *JAG2* variants that predict protein loss of function and satellite cell depletion via a decrease in Pax7+ cells. Additionally, downregulation of *JAG2* in murine myoblasts resulted in downregulation of several molecules within the Notch pathway, implying a disease mechanism correlated to defects in the Notch pathway. Table 2 summarizes the articles discussed above.

#### 3.3.2. Wnt

Wnt signaling regulates various biological processes, including cell migration, polarity, and motility, and is critical for embryonic formation and development as well as tissue homeostasis in adults [66,67]. Wnt signaling is a pleiotropic signal that regulates various biological processes, including cell migration, polarity, and motility. It is critical for embryonic formation and development, as well as adult tissue homeostasis [66,67,68,69].

Concisely, Wnt signaling diverges into canonical and non-canonical pathways. The canonical β-catenin pathway, also known as the Wnt/β-catenin pathway, regulates cell proliferation by activating target genes in the nucleus via TCF/LEF protein. The non-canonical pathway branches further into two main pathways: 1. the planar cell polarity pathway that coordinates cellular assembly by controlling cell migration, polarity, and asymmetric division, and 2. a calcium-dependent pathway, which is also known as Wnt/Ca^2+^ pathway [66,67,69,70].

Fujimaki et al. in their 2014 report, investigated the Wnt signaling pathway activation in mice models using voluntary wheel running as a mild physiological stimulus. In contrast to aged skeletal muscle, triggering exercise-dependent canonical Wnt/β-catenin signaling did not cause fibrosis in adult skeletal muscle, was associated with satellite cell activation, and promoted myogenic proliferation. In young mice, voluntary exercise significantly restored *Pax7* expression in satellite cells, whereas in adult mice the decrease in *Pax7* due to aging was primarily restored by the activated Pax7 cell population. Interestingly, Murphy et al. proposed that Wnt/β-catenin signaling must be silenced during adult muscle regeneration, because transiently active Wnt/β-catenin signaling is not required for regeneration and β-catenin downregulation is necessary to restrict the regenerative response, but does not participate in self-renewal of the satellite cell. MyoD+ cell expansion and proliferation were unaffected by *β-catenin* deletion in the adult mice model, indicating that satellite cells could self-renew, as assessed by Pax7+ cell number in the basal lamina.

*Klotho* is a Wnt signaling negative regulator that is often associated with aging [71,72]. *Klotho* deficiency in mice has been linked to muscle weakness, sarcopenia, and compromised skeletal muscle regeneration [73,74,75]. Additionally, studies suggest that *Klotho* plays a role in modulating autophagy in different type of tissues [76]. Wehling-Henricks et al. discovered that *Klotho* expression in muscle is reduced in mdx dystrophic mice at the onset of pathology and during aging. The dramatic production of pro-inflammatory cytokines like tumor-necrosis factor α (*TNFα*) and interferon gamma (*IFNγ*) in mdx mice reduced *Klotho* expression in mdx dystrophic muscle, while elevating *Klotho* expression in macrophages by 50% in each case, suggesting it may regulate cytokine expression in dystrophic muscle. *Klotho* regulates satellite cell numbers and muscle fiber growth in vivo, as exhibited by mdx mice that received the *Klotho* transgenic bone marrow cells (BMCs). The increased muscle fiber size and quantification of Pax7+ satellite cells in recipient mice compared to the mdx control, suggested an increase in the number of macrophages that promote muscle regeneration. Wehling-Henricks et al. suggested that *Klotho* enhances the regenerative capacity of dystrophic muscle driven by *TNFα*. *Klotho* suppression is an adaptive trait that shields macrophages from oxidative stress damage. M2 macrophages can be beneficial or harmful in muscular dystrophy, but administrating *Klotho* to dystrophic muscle can reduce the expression of pro-fibrotic transcripts and prevent the connective tissue accumulation.

Alexander et al. used a different strategy to understand differentiation regulator factor change within Wnt signaling by profiling commonly dysregulated miRNAs in a zebrafish dystrophic model (sapje zebrafish) and human DMD. They revealed that several muscle-enriched miRNAs, including miR-199a-5p, were more highly expressed in sapje zebrafish at 5 days post fertilization (dpf) than at 30 dpf, indicating that dysregulated miRNAs in zebrafish dystrophic disease may arise early in development. They also discovered that miR-199a-5p level were elevated in human dystrophic muscles. Serum response factor (SRF) is a miRNAs repressor, and its transcriptional co-factors myocardin-related transcription factor (MRTF) were used in this research to identify dysregulated miRNA transcripts. They are known to control myogenic fusion, differentiation, regeneration, and dystrophic disease pathogenesis [77,78,79]. MiR-199a-5p expression is SRF-dependent and targets several regulatory factors within Wnt signaling pathways including *Wnt2*, Frizzled-4 (*FZD4*), and Jagged1 (*JAG1*) which are involved in human myoblast differentiation, regulation of *Pax7/MyoD* expression, and proliferation. The reciprocal regulation of miR-199a-5p with these regulatory factors was discovered to be essential in diagnosing DMD at an early stage via a non-invasive diagnostic approach using blood serum from young patients [80]. Therefore, regulating miR-199a-5p may have a beneficial impact on myogenic progression to ameliorate dystrophic skeletal muscle. The findings of the mentioned articles are presented in Table 3.

#### 3.3.3. Alternative Signaling Pathways

Ganassi et al. explored the activation of mTORC1 signaling in muscle stem cells in *MyoG* null zebrafish model. The mechanistic target of the rapamycin (mTOR) signaling pathway is essential for cellular growth and metabolism. It has two distinct complexes, mTORC1 and mTORC2, which interact with various proteins. mTORC1 is activated by nutrients, while mTORC2 is regulated by phosphoinositide 3-kinases (*PI3K*) and growth factor signaling [81,82]. Adult zebrafish with *MyoG* loss have reduced muscle bulk, implying a continuous muscle defect in adulthood. MyoG−/− zebrafish mutants have more abundant muscle stem cells than their siblings, with an over thirteenfold increase in myofiber nuclei. *MyoG* regulates the adult skeletal muscle growth rate and muscle stem cell dynamics, by affecting their number and deep quiescence state, as well as regulating the expression of mTORC1 signaling genes. The absence of *MyoG* raised mTORC1 signaling in adult muscle, indicating an alteration in the muscle stem cell niche, and resulted in precocious muscle stem cell activation as assessed by increased *Pax7* mRNA expression, cell number, and proliferation potential. However, by downregulating the fusogenic genes, *Myomixer* and *Myomaker*, the stem cells were unable to fuse, and produced improper myofibers with increased size but decreased length, indicating early terminal differentiation.

Hindi and Kumar reported that *TRAF6* and *Pax7* are highly expressed in quiescent and activated satellite cells. Satellite-cell specific *TRAF6* knock-out mice (TRAF6^scko^) exhibited a decrease in *Pax7* with the loss of quiescence satellite cell population, as well as upregulation of *MyoD* expression in cultured myogenic cells, resulting in precocious differentiation. Tumor necrosis factor receptor-associated factor 6 (*TRAF6*) is a protein involved in mediating signal transduction pathways, including extracellular signal-regulated kinase (ERK), Jun N-terminal kinase (JNK) and p38 mitogen-activated protein kinase (MAPK). *TRAF6* has a pivotal role in regulating *Pax7* expression in satellite cells, as observed by a decrease in the number of activated satellite cells, caused by an intrinsic defect. A substantial decrease in quiescent Pax7+ satellite cells in newly isolated fibers of TRAF6^scko^ mice was observed, suggesting its importance for the maintenance of the satellite stem cell pool and function in vivo. TRAF6^scko^ mice had reduced muscle regeneration after injury using a BaCl_2_ injection, while defective muscle repair was still observed after weeks post-injury. Using wild-type *TRAF6* and TRAF6C70A mutant mice, Hindi and Kumar examined the activity of *TRAF6* E3 ubiquitin ligase in satellite cell proliferation and self-renewal. It was discovered that the activity of *TRAF6* E3 ubiquitin ligase is critical for self-renewal, proliferation, and preventing precocious differentiation. Despite the fact that *TRAF6* is crucial for *AKT* phosphorylation and translocation, and for response to growth stimuli, no significant difference was observed in *AKT* phosphorylation or total levels and in its downstream phosphorylation targets. In order to proliferate and self-renew, satellite cells require the ERK1/2 and JNK1/2 signaling activation, while *c-JUN* regulates *Pax7* expression by directly binding to the *Pax7* promoter. *TRAF6* deficiency caused the E3 ubiquitin ligases to function improperly, resulting in enhanced *c-JUN* degradation. *TRAF6* deletion in satellite cells inhibits regeneration and intensifies myopathy, suggesting a post-transcriptional role for the MAPK/c-JUN/AP1 axis in mediating *Pax7* expression. Increased p38 MAPK expression in TRAF6-deficient cells promotes precocious differentiation, while decreased ERK1/2 and JNK1/2 signaling promote an exit from quiescence and precocious differentiation.

Ogura et al. investigated the role of *TGF-β* activated kinase 1 (*TAK1*) in dystrophic muscle, which is necessary for myofiber regeneration after injury and is activated in adult satellite stem cells. *TAK1* was reported to be activated in response to *TRAF6* polyubiquitination, which then engages the Nuclear Factor kappa-light-chain-enhancer of Activated B cells (NF-κB) and the MAPK signaling pathways [83,84]. NF-κB is one of the important signaling pathways activated through *TAK1*-dependent mechanisms. They revealed that satellite-cell-specific *TAK1* knock-out mice (TAK1^scko^) had reduced muscle regeneration, with a decrease in the number and size of centrally nucleated fibers as well as cell arrest with the reduction in the G2/M phase in the cell cycle. *TAK1* inactivation inhibited regenerative myogenesis in mice by reducing fiber size and the number of myogenic factors (embryonic-Myosin Heavy Chin (*eMyHC*), *Myf5*, *MyoD* and *MyoG* in both injured and uninjured mice models, indicating that myogenesis was reduced during early regeneration. *TAK1* is necessary for preserving the satellite stem cell pool in the skeletal muscle of adults, and its inactivation compromises muscle regenerative capability, which causes muscle wasting. The Pax7−/MyoD+ cell population significantly increased in TAK1^scko^ mice, while the Pax7+/MyoD+ activated stem cell population decreased, explaining that *TAK1* is essential for satellite cell generation, proliferation, and self-renewal; however, it does not regulate *MyoD* expression while inhibiting premature progression in the myogenic cell lineage. Moreover, *TAK1* is crucial for satellite cell survival, and *TAK1* disruption can result in necroptosis. The increase in reactive oxygen species (ROS) in TAK1^scko^ mice was shown by the rescue of the majority of cell deaths, Pax7+ proliferation, and *MyoD* expression in satellite cells, by the necroptosis inhibitor. Therefore, *TAK1* inactivation inhibits NF-κB pathways in satellite cells, leading to poor survival and proliferation, and it is essential for satellite cell proliferation and survival, as well as regenerative myogenesis. Additionally, *TAK1* is the key for cellular homeostasis, and reduced proliferation of satellite cells in TAK1^scko^ mice may be due to the diminished sustainability and increased mortality of satellite cells. Table 4 compiles the findings of the selected articles discussed above.

## 4. Conclusions

Overall, *Pax7* plays a critical role in muscle development, and contributes to skeletal muscle repair through its regenerative function. In terms of signaling pathways, research on *Pax7* expression levels indicates its interaction with the Notch and Wnt pathways during myogenesis. This interaction is vital for regulating the cell fate and behavior of muscle stem cells, underscoring the complex regulatory network involved in muscle formation and repair. However, when it comes to muscular dystrophy, there is a lack of information regarding *Pax7*’s specific role at various stages of the disease progression. Investigating *Pax7*’s function at these specific time-points could prove to be a fascinating area of study, potentially revealing whether modulating *Pax7* expression levels at these stages would impact the severity and progression of the disease. By gaining a deeper understanding of how *Pax7* interacts with other signaling pathways, we hope that future research can uncover the underlying mechanisms involved in myogenic programming. This insight could open new avenues for manipulating *Pax7* expression, leading to potential therapeutic strategies for treating muscle-related conditions.

Realistically, relying solely on *Pax7* as a therapeutic target in muscular dystrophy treatment would be challenging, due to the diverse forms of muscular dystrophy and their respective genetic mutations. Muscular dystrophies, despite their different underlying causes, share a common characteristic of progressive muscle wasting, in which *Pax7* could play a crucial role. Utilizing *Pax7*’s regulatory function could be advantageous, particularly in combination with existing approaches in muscular dystrophy therapy. In the case of dystrophinopathies, *Pax7* cannot address the lack or complete absence of *dystrophin*. However, it could be used as a complementary approach to support *dystrophin* rescue efforts. Similarly, in Facioscapulohumeral muscular dystrophy, inducing *Pax7* overexpression in regions where it is repressed might hold the potential to ameliorate disease pathology [85]. Additionally, while *Pax7* shows promise as a valuable player in mitigating muscle wasting, it is more practical to view it as part of a comprehensive approach in combination with other therapeutic strategies, for addressing the complexities of muscular dystrophy effectively. By understanding and harnessing *Pax7’*s regulatory potential, we can enhance current therapeutic efforts and pave the way for more effective treatments for various forms of muscular dystrophy.

Furthermore, it would be essential to investigate whether muscular dystrophy models undergoing treatment via pharmacological strategies could benefit from the regenerative potential of *Pax7* overexpression when induced. Understanding the mechanistic and synergistic effects of *Pax7* in combination with existing therapies could potentially refine and complement current therapeutic strategies. Ultimately, this could alter the direction and dynamics of the treatment method for muscular dystrophy, and a hopeful solution brought from the bench to bedside.

## Figures and Tables

**Figure 1 ijms-24-13051-f001:**
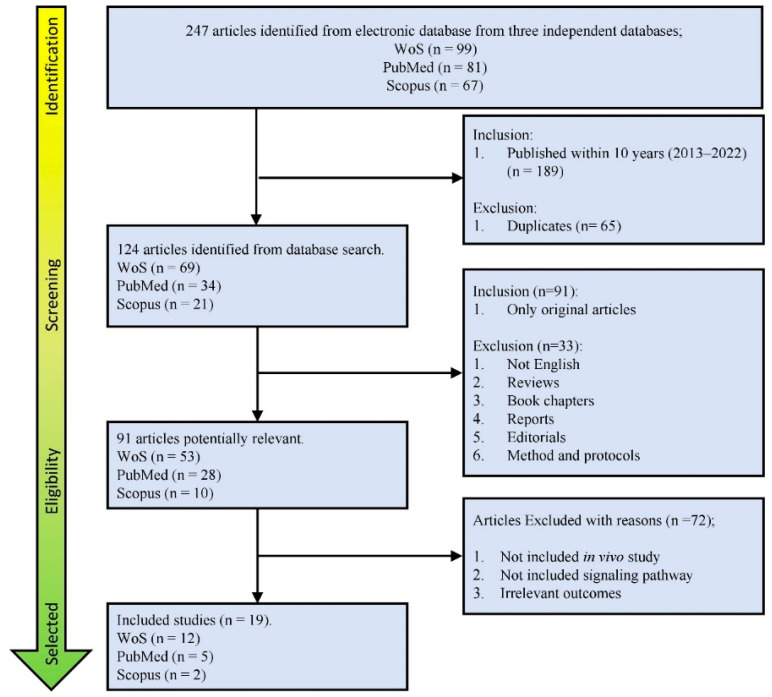
Flowchart presenting the article identification through three independent electronic databases (Web of Science (WoS), PubMed and Scopus), which resulted in a total of 19 articles being included in the final discussion.

**Table 1 ijms-24-13051-t001:** Tabulation of risk-of-bias analysis.

Risk bias domain		[21]	[22]	[23]	[24]	[25]	[26]	[27]	[28]	[29]	[30]	[31]	[32]	[33]	[34]	[35]	[36]	[37]	[38]	[39]
**++**	Definitely low risk of bias																			
+	Probably low risk of bias																			
−/NR	Probably high risk of bias																			
--	Definitely high risk of bias																			
N/A	Not applicable																			
Clear hypothesis/objectives		++	++	++	++	++	++	++	++	++	++	++	++	++	++	++	++	++	++	++
Interventions clearly described		++	++	++	++	++	++	++	++	++	++	++	++	++	++	++	++	++	++	++
Was the administered dose or exposure level adequately randomized?		N/A	N/A	N/A	N/A	N/A	N/A	N/A	N/A	N/A	N/A	N/A	N/A	N/A	N/A	N/A	N/A	N/A	N/A	N/A
Was allocation to study groups adequately concealed?		N/A	N/A	N/A	N/A	N/A	N/A	N/A	N/A	N/A	N/A	N/A	N/A	N/A	N/A	N/A	N/A	N/A	++	N/A
Were experimental conditions identical across study groups?		N/A	++	++	++	++	++	++	++	++	++	++	++	++	++	++	++	++	++	++
Can we be confident in the exposure characterization?		++	++	++	++	++	++	++	++	++	++	++	++	++	++	++	++	++	++	++
Were the research personnel and human subjects blinded to the study group during the study?		N/A	N/A	N/A	N/A	N/A	N/A	N/A	N/A	N/A	N/A	N/A	N/A	N/A	N/A	N/A	N/A	N/A	++	N/A
Were the outcome data complete without attrition or exclusion from analysis?		**++**	**++**	**++**	**++**	**++**	**++**	**++**	**++**	**++**	**++**	**++**	**++**	**++**	**++**	**++**	**++**	**++**	**++**	**++**
Can we be confident in the outcome assessment?		++	++	++	++	++	++	++	++	++	++	++	++	++	++	++	++	++	++	++
Were all measured outcomes reported?		++	++	++	++	++	++	++	++	++	++	++	++	++	++	++	++	++	++	++
Were n, N and statistical methods appropriate?		++	++	++	++	++	++	++	++	+	++	++	++	++	++	++	++	++	++	++
Were adverse events reported?		N/A	N/A	N/A	N/A	N/A	N/A	N/A	N/A	N/A	N/A	N/A	N/A	N/A	N/A	N/A	N/A	N/A	N/A	N/A

**Table 2. ijms-24-13051-t002:** Summary of selected articles in Notch signaling.

No.	Author(s)	Year of Publication	Animal Model	Key Findings
1	Coppens et al. [21]	2021	*UAS-ds drpr line* drosophila, *Ser-Gal4* driver line drosophila, transgenics *UAS-ds drpr* transgene with *Ser-Gal4* driver flies were crossed to generate RNAi flies that downregulate Drpr in *Serrate*-positive cells, standard genetic background strain flies with *Ser- Gal4* flies were crossed as control progeny	1. The study focused on a form of muscular dystrophy that is associated with pathogenic variants in *JAG2*. 2. *JAG2*-related muscular dystrophy associated with: Depletion of satellite cells. Reduction in Pax7+ cells. 3. Downregulation of *JAG2* in murine myoblasts led to: Downregulation of several molecules in Notch pathway, including *Pax7*.
2	Fiore et al. [22]	2020	PKCθ−/− (C57BL/6J background), mdx (C57BL/10ScSn-Dmdmdx/J), mdx θ−/− transgenic (C57BL/6j-C57BL/10ScSn background) and C57BL/10ScSn control mice	1. The study demonstrated that the lack of *PKCθ* promotes the regenerative ability of muscle stem cells in the context of chronic muscle injury. 2. Loss of *PKCθ*: Enhances regeneration and muscle repair in dystrophic muscle. Preserves satellite cell pool by expressing higher Pax7+ cells. Increases *Notch1* expression. Reduces muscle fiber degeneration compared to dystrophic mice control.
3	Fujimaki et al. [23]	2018	*Notch1*-floxed, *Notch2*-floxed, Pax7^CreERT2^ mice were crossed with floxed mice to generate Pax7^CreERT2/+^;Notch1(N1)^f/f^, Pax7^CreERT2/+^;Notch2(N2)^f/f^, and Pax7^CreERT2/+^;N1^f/f^;N2^f/f^ mice	1. The study revealed that *Notch1* and *Notch2* coordinately regulate the function of satellite cells in both quiescent and activated states. Maintain the stem cell pool in quiescent state. Prevent premature activation of stem cells. Influence stem cell fate decisions. 2. Pax7+ satellite cell number per myofiber: Reduced significantly in N2-scKO. Almost eliminated in DKO mice. 3. Notch1 and Notch2 have compensatory effects: Work together to sustain the satellite cell numbers in adult muscle.
4	Pisconti et al. [24]	2016	*Sdc3*^−/−^ mice, mdx^4cv^ mice, wild type and mdx^4cv^;*Sdc3*^+/+^ controls. The double mutant mdx^4cv^;*Sdc3*^−/−^ mice (C57Bl/6 background)	1. The study showed that the loss of niche-satellite cell interactions in *Sdc3* null mice alters muscle progenitor cell homeostasis, leading to improved muscle regeneration. 2. Ablation of *Sdc3* resulted in the depletion of Pax7+ satellite cells after injury in both dystrophic and non-dystrophic mice. 3. Despite the loss of Pax7+ cells, *Sdc3* null mice showed improved muscle regenerative capacity and reduced fibrosis. 4. The increase in myogenin+ cells suggests that the depletion of Pax7+ satellite cells in *Sdc3* null mice did not cause the regeneration exhaustion, even in aged mice.
5	Servián-Morilla et al. [25]	2016	Drosophila transgenes expressing human *POGLUT1*^wt^ (wild type) and human *POGLUT1*^D233E^. *w*^1118^ (wt), *rumi*^79/79^, *Mef2-GAL4/UAS attB-POGLUT1*^WT^-*FLAG*; *rumi*^79/79^, and *Mef2-GAL4/UAS attB-POGLUT*^D233E^-*FLAG*; *rumi*^79/79^, *Mef2-GAL4* strain	1. The study demonstrated that a *POGLUT1* mutation leads to a muscular dystrophy characterized by reduced Notch signaling and loss of satellite cells. 2. *POGLUT1* with D233E mutation leads to: Partial defect in α-dystroglycan glycosylation. Reduction in Notch signaling. Alteration in myogenesis. 3. Overexpression of NICD rescued the above defects: Increased the population of differentiating cells. 4. However, NICD overexpression also resulted in: Exhaustion of quiescent Pax7+ pool.
6	Bi et al. [26]	2016	*Myl1*^Cre^, Ckmm-Cre (MCK-Cre), *Rosa*26(Gt)Sor^N1ICD^, *Rosa*26 (Gt)Sort^d-Tomato^, *Rosa*26(Gt)Sor^nTnG^, mdx, and *CpGFP* mice.	1. The study explored the stage-specific effects of Notch activation during skeletal myogenesis. 2. *Notch1* activation can revert de-differentiated myocytes back into Pax7+ quiescent stem cells. 3. In MLC-N1ICD mice: High expression of Pax7+ cells. 4. In MCK-N1ICD mice: No de-differentiation. *Notch1* activation and enhanced muscle function and regeneration in aged mice. 5. *Notch1* activation is stage-dependent in post-differentiation muscle cells. 6. It improves myotube’s role as a stem cell niche, supporting muscle regeneration and function.
7	Jiang et al. [27]	2014	*Myf5*^nLacZ^, *ROSA*26^N1ICD^, mdx, Pax7^CreER^, and *Cp-GFP* mice	1. The study revealed that age-dependent depletion of satellite cells in muscular dystrophy is attributed to Notch signaling deficiency. 2. Mdx mice expressed higher *Pax7* levels compared to wild type, at all ages. 3. Young mdx mice showed down-regulated Notch markers. 4. Impaired Notch signaling affects the ability of cells to self-renew and revert to a quiescent state in aged mice. 5. Notch inhibits *MyoD* and *myogenin*, leading to an absence of myogenic differentiation. 6. Notch activation improves Pax7+ expression, suggesting that fine-tuning of Notch signaling is essential in myogenesis.
8	George et al. [28]	2013	*Numblike* (*Nbl*^−/−^), conditional *Numb* mutant (Numb^fl/fl^), *Pax7*-*Cre* (*Pax7*^ICNm^), R26R^YFP^, *Numb*^tm1Zili/tm1Zili^/J, *Numb*^ltm1Zili/tm1Zili^/J, and the TMX-inducible Cre recombinase line *CAGG ER-Cre* mice	1. The study showed that Numb-deficient satellite cells exhibit defects in regeneration and proliferation. 2. Absence of Numb in the Pax7-lineage: Reduced satellite cells on single-fiber culture. Causes a defective repair response in adult muscle. 3. Numb deficiency does not substantially alter expression levels of *MyoD*, *Pax7* and *Notch* ligands in satellite cells. 4. Notch signaling is not the primary mechanism regulating satellite cell proliferation. 5. Silencing *Myostatin* rescues proliferation in Numb-deficient mice.

**Table 3. ijms-24-13051-t003:** Summary of selected articles in Wnt signaling.

No.	Author(s)	Year of Publication	Animal Model	Key Findings
1	Wehling-Henricks et al. [29]	2018	C57BL/10ScSn-Dmd^mdx^/J (mdx mice), B10.129P2(B6)-Il10^tm1Cgn^/J (*IL10* mutant mice), B6.129S-Tnf^tm1Gkl^/J (*TNFa* mutant mice) and C57BL/6 (wild-type mice)	1. The study demonstrated that in the mdx mouse model, macrophages contribute in muscle growth through a *Klotho*-mediated pathway. 2. Elevated *Klotho* expression from macrophage populations: Induces an M2 macrophage phenotype. Enhances regeneration capacity. 3. This enhancement is achieved by: Increasing satellite cell number. Promoting satellite cell proliferation and growth. 4. Mdx-recipient mice receiving *Klotho* transgenic BMCs showed: Increased Pax7+ satellite cells. Elevated muscle fiber size. 5. In contrast, mdx-recipient mice receiving wild-type BMCs did not exhibit these improvements.
2	Fujimaki et al. [30]	2014	C57BL/6J mice	1. The study demonstrated that Wnt protein-mediated satellite cell conversion occurs in both adult and aged mice, following voluntary wheel running. 2. Voluntary exercise has beneficial effects on Pax7+ expression in satellite cells: Particularly prominent in young mice. Restores the decrease in *Pax7* in aging adult mice. 3. *Notch* ligand expression slightly decreased after exercise. 4. Wnt/β-catenin signaling pathway activation is the major mechanism controlling satellite cell activation. 5. Low exercise intensity is effective in preventing the progression of sarcopenia in aged skeletal muscle.
3	Murphy et al. [31]	2014	*Pax7*^CreERT2^, *β-catenin*^fl2−6^, *β-catenin*^fl3^, *Rosa*^mTmG^, and *TCF/Lef*:*H2B-GFP*^Tg^ mice, all in C57/BL6J background	1. The study concluded that transiently active Wnt/β-Catenin signaling is not required, but it must be silenced for proper stem cell function during muscle regeneration. 2. Activation of Wnt/β-Catenin signaling in myogenic cells occurs during muscle regeneration. 3. However, *β-Catenin* is not required within satellite cells for muscle regeneration or satellite cell self-renewal, based on the number of Pax7+ cells in the basal lamina. 4. Down-regulation of *β-Catenin* is critical in adult muscle for: Limiting the prolonged regenerative response of myoblasts. Preventing immature differentiation. Reducing fibrosis. 5. In fetal myogenesis, *β-Catenin* is required for: Regulation of fetal myofiber differentiation. Determining cell fate. Expansion of *Pax7*.
4	Alexander et al. [32]	2013	Wild-type (AB strain) zebrafish, the *sapje* fish line, *Tg*(*mylz2- miR-199a-5p*) zebrafish, wild-type C57Bl6/J and mdx^5cv^ mice	1. The study showed that miR-199a-5p is induced in dystrophic muscle and influences Wnt signaling, cell proliferation, and myogenic differentiation. 2. Early-dpf sapje zebrafish showed high expression of several muscle-enriched miRNAs. 3. Overexpression of miR-199a-5p did not affect normal myoblast differentiation into myotubes. 4. Inhibition of miR-199a-5p in normal myoblasts caused: Decreased myogenic differentiation. Increased levels of MyoD1 and Pax7 protein. Enhanced proliferation. Upregulation of endogenous Wnt signaling factors. Reduced *β-Catenin* expression.

**Table 4. ijms-24-13051-t004:** Summary of the selected articles in the alternative signaling pathways.

No.	Author(s)	Year of Publication	Animal Model	Signaling Pathways	Key Findings
1	Ganassi et al. [33]	2020	myog^fh265^ mutant allele, Myog^kg125^(myog−/−), TgBAC(pax7a:GFP)^t32239Tg^, and myog^kg125^;TgBAC(pax7a:GFP)^t32239Tg^ zebrafish	mTORC1 signaling	1. The study confirmed that *MyoG* is a crucial regulator of adult myofiber growth and muscle stem cell homeostasis. 2. Loss of *MyoG* causes mTORC1 signaling activation in muscle stem cells, leading to: Increased *Pax7* expression and cell number. Downregulation of fusogenic genes. Improper myofiber growth in zebrafish. 3. *MyoG* knockout in mice: Leads to *MyF* accumulation in neonatal limb muscle. Increases Pax7+ cell population. 4. *MyoG* plays an essential role in tissue homeostasis by controlling the balance between the quiescence and activation of satellite cells.
2	Nunes et al. [34]	2017	dy^W^, cross between mutants and wild-type (WT) controls to generate heterozygous dy^W^ mice	JAK-STAT signaling	1. This study indicated that impaired fetal muscle development and JAK-STAT activation are key markers of disease onset and progression. 2. *Laminin 211* rescues muscle growth by: Attenuating STAT3 signaling. 3. Increased JAK-STAT signaling in dy(^W−/−^): Over-activates *MyoD*. Causes premature cell cycle exit. Leads to the formation of fewer differentiated and fusion-competent cells. 4. dy(^W−/−^) pups exhibit an aging muscle phenotype. 5. They cannot recover during fetal development, due to: Low *Pax7* expression. Impaired proliferation. Low level of self-renewal capacity.
3	Hindi & Kumar [35]	2016	*Pax7-CreER* were crossed with *Traf6*^fl/fl^ to generate TRAF6^scko^ mice, mdx were crossed with TRAF6*scko* to generate mdx;TRAF6^scko^ mice and littermate mdx;Traf6^fl/fl^ mice, TRAF6^scko^ were crossed with mT/mG (Gt(ROSA)26Sor^tm4(ACTB–tdTomato,–EGFP)Luo^/J) mice	ERK1/2-JNK-c-JUN signaling	1. The study demonstrated that *TRAF6* plays a critical role in regulating satellite cell self-renewal and function during regenerative myogenesis. 2. *TRAF6* knockout mice showed: Downregulation of *Pax7* expression. Upregulation of *MyoD*, leading to precocious differentiation. 3. *TRAF6* is required for the replicative capacity of satellite cells. 4. In *TRAF6* knockout mice, satellite cells exhibiting changes in signaling pathways: Downregulated ERK and JNK signaling. Upregulated p38 MAPK signaling. 5. *TRAF6*-null primary myogenic cells exhibited a reduction level in c-JUN protein, both total and phosphorylated. 6. *TRAF6* and *c-Jun* are modulators of *Pax7* expression.
4	Farini et al. [36]	2016	Mdx (C57BL/10 background, C57BL/10-mdx), and C57BL/6J mice	PLC/IP3R/Ca^2+^/PKCα signaling	1. The study showed that delayed myogenesis in DMD fetal muscle is mediated by inositol 1,4,5-triphosphate (IP3)-dependent Ca^2+^ signaling. 2. No signs of fiber degeneration or apoptotic activation are observed in dystrophic fetal muscles. 3. Regenerating myofiber numbers are not altered in dystrophic fetal muscles. 4. DMD myofiber exhibits significantly higher Pax7+ cell count compared to healthy, 12-week-old human fetal muscle samples. 5. The absence of dystrophin in DMD myofiber can lead to dysfunction of calcium influxes and releases. 6. This dysfunction has cumulative consequences for fetal DMD myogenesis, and may contribute to adult muscle damage.
5	Tierney et al. [37]	2016	*Pax7CreER^TM^*, *Pax7CreER*^TM^;*R26R*^TdT^, *luciferase*, *EGFP*, C57BL/6, *NOD/SCID*, and *R26R*^TdT^ mice	Notch- and canonical Wnt-signaling	1. The study revealed that autonomous extracellular matrix remodeling plays a critical role in controlling a progressive adaptation in satellite cells’ regenerative ability during development. 2. Fetal muscle stem cells adapt to their microenvironment by: Autonomously secreting ECM. Supporting and enhancing stem cell function intrinsically. Maintaining *Pax7* expression. Decreasing *Myf5* and *MyoG* expressions. 3. Fetal muscle stem cells have: Higher levels of Notch targets. Lower levels of canonical Wnt targets than adult activated muscle stem cells. 4. Co-transplantation of fetal and adult muscle stem cells: Benefits adult muscle stem cells. Increases *Pax7* expression in adult muscle stem cells. Influences ECM molecule remodeling. Improves adult muscle stem cells’ regenerative potential.
6	Ogura et al. [38]	2015	*Pax7CreER* crossed with TAK1^f/f^ mice to generate TAK1^scko^ mice	NF-kB and JNK signaling	1. The study demonstrated that *TAK1* plays a crucial role in modulating satellite cell homeostasis and skeletal muscle repair. 2. Pax7+ expression was reduced in *TAK1* knockdown mice upon injury, resulting in satellite cell pool depletion, and reduced Notch signaling. 3. TAK1scko mice showed: Reduction in cell cycle G2/M phase. Decreased activated stem cells. Induction of differentiated cells. 4. *TAK1* is required for *Pax7* maintenance in satellite cells but not for *MyoD* expression and preventing precocious progression in myogenic lineage. 5. ROS levels were increased in TAK1^scko^ and caused necroptosis and apoptosis. 6. *TAK1* positively modulates the NFk-β signaling pathway, rescuing ROS-induced necroptosis 7. Reconstitution of the JNK signaling pathway rescued proliferation in TAK1^scko^.
7	Mu et al. [39]	2015	Wild-type (C57BL/10J background), mdx (dys^−/−^) and dKO (dys^−/−^;utrn^−/−^) mice	Notch and TGF-β signaling	1. The study elucidated the role of Notch/TGF-β signaling in muscle progenitor cell depletion and its contribution to the rapid onset of histopathology in muscular dystrophy. 2. Aged mice showed reduction in *Pax7* expression and proliferation activity, compared to younger mice. 3. dKO mice have reduced muscle progenitor cells (MPC) and Pax7+ cells, compared to mdx mice. 4. The reduction in MPC and Pax7+ cells in dKO mice accelerates cell senescence, fibrosis, and adipose tissue formation, when compared to mdx mice. 5. dKO mice show overexpression of Notch signaling, TGF-β signaling and pro-inflammatory factors, compared to mdx. 6. Overexpressed of Notch negatively regulates muscular dystrophy phenotypes more effectively in mdx model compared to dKO mice model.

## Data Availability

The study ogininal and reported data are available within the article and Appendix A. For additional inquiries, kindly direct them to the corressponding author.

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
