# Peer review of "PAX7, a Key for Myogenesis Modulation in Muscular Dystrophies through Multiple Signaling Pathways: A Systematic Review"

_ijms, 2023, doi:10.3390/ijms241713051_

Round 1

Reviewer 1 Report

The authors have performed a very detailed review on the molecular pathways involved in myogenesis highlighting the key role of Pax7.

The English style has to be reviewed extensively.

Author Response

Dear Reviewer 1.

Reviewer 2 Report

Summary

This manuscript is a review of the expression, function, and signaling of Pax7 in muscular dystrophy based on a literature search of 19 eligible articles. Although the previous results are well reviewed, it is unsatisfactory for future prospects. In particular, Figure 2 and Conclusion need to be significantly revised as noted below to improve the quality of this study for publication in the International Journal of Molecular Sciences.

Comments

1.          Line 17: “PRISMA” should be defined.

2.          Line 40: “disease severity and”???

3.          Line 58: “Myogenic Differentiation 1 (MyoD)” should be described in Line 54.

4.          Line 132: “Myogenic Differentiation 1 (MyoD)” should be “MyoD”.

5.          Lines 193 253: “dyw mice model” and “mdx mice model” need to be briefly introduced for the readers who are not familiar with this field.

6.          Line 238: Underline of “MyoG” should be deleted.

7.          Table 2: No need to segment the table, they must be one table. The column of “Signaling pathways” can be deleted because all the articles in Table 2 describe Notch signaling.

8.          Table 3: It should be revised in the same way as Table 2.

9.          Line 371: The direct targets of miR-199a-5p and its relationship to DMD should be presented and cited.

10.          Table 4: It should be revised in the same way as Table 2.

11.          Figure 2: The level of Pax7 expression in quiescent satellite cells should be illustrated. Figure 2 illustrates only the healthy state. It does not show at all the roles, effects, or functions of Pax7 in muscular dystrophy, which is the main topic of this manuscript. Figure 2 needs to be substantially revised and improved.

12.          This manuscript reviewed the previous articles well, but did not provide novel and original insights from the authors. Discuss the scientific importance, biological significance, or therapeutic applicability of Pax7 studies in muscular dystrophy, especially in Conclusion.

Minor errors should be proofed.

Author Response

Dear Reviewer 2.

Reviewer 3 Report

The article is poorly constructed and does not specify what it brings new compared to other reviews, which were not even taken into account. The realization scheme is without head and tail. English is deficient and confusing in many places. Many explanations are missing. Not all the selected articles refer to muscular dystrophies.

Title: a comma should be placed after Pax7, not a semicolon.

This review article has a very unusual and inadequate structure, with Materials and Methods and Results. The article (in this form) is expected to be about how to make a choice of articles for a review. The explanations for study selection and data collection are far too broad. There is no explanation why articles with in vitro experiments or review articles were not taken into account.

It is known that muscular dystrophies are genetic diseases and even if myogenesis, through Pax7 regulation, would be resolved positively, the newly formed muscle fibers still remain without the involved protein. But maintaining muscle mass for a longer or shorter time could have a therapeutic impact. It could have been an interesting update for the chosen subject, if it had been treated with more attention.

There are some phrases that are difficult to understand or interpret because the expression in English is cumbersome and inadequate. E.g. “Muscular dystrophy resulting from mutations occurs in the genes responsible for muscle system development and interrupts the normal mechanisms of myogenesis…”; ”…a gene responsible for providing instructions for making protein…”; “ Wnt signaling occurs via two pathways; noncanonical, which is independent of β-catenin…..” (in fact, there are three signaling pathways, canonical, noncanonical planar cell polarity and noncanonical Wnt/calcium pathway; a normal expression must show that the canonical one involves beta-catenin, while the non-canonical ones activate independently of beta-catenin. The expression of the authors is ambiguous.) etc.  

There are not 9 forms of muscular dystrophy, but 9 types, each comprising several forms.

Subsection 3.1. it should have been part of the introduction.

Articles by Ganassi et al, Tierney et al., Fujimaki et al, 2014, Murphy et al., do not refer to muscular dystrophies.

Subchapter 3.3.3 is called: Alternatives. Alternatives to what? There is no explanation. In the title of Table 4, "alternative signaling pathways" appears.

Figure 2 appears in the text only in Conclusion, but is not explained in detail anywhere, neither in the text nor under the figure.

There are some phrases that are difficult to understand or interpret because the expression in English is cumbersome and inadequate. E.g. “Muscular dystrophy resulting from mutations occurs in the genes responsible for muscle system development and interrupts the normal mechanisms of myogenesis…”; ”…a gene responsible for providing instructions for making protein…”; “ Wnt signaling occurs via two pathways; noncanonical, which is independent of β-catenin…..” (in fact, there are three signaling pathways, canonical, noncanonical planar cell polarity and noncanonical Wnt/calcium pathway; a normal expression must show that the canonical one involves beta-catenin, while the non-canonical ones activate independently of beta-catenin. The expression of the authors is ambiguous.) etc.  

Author Response

Dear Reviewer 3.

Reviewer 4 Report

This review of Pax7 in muscle development, regeneration and in certain models of muscular dystrophy was approached from a very broad view and incorporated a clear inclusion/exclusion criteria.

Some issues that would improve this review are listed below:

1. The extensive discussion of how papers were chosen is not needed. Retaining fig.1 and a succinct description of their criteria will be sufficient.

2.Tables 2-4 provide a nice source of information for the reader, but the key findings should be discussed in the text, not the tables.

3.The authors often point out a gene as a target for MD therapeutics, but as dystrophies are caused by many genes with different etiologies there should be more discussion on why this would be a target and for which genetic mutations.

4. Generally there is a lack of details in the descriptions, What is the model system?  Is it protein or mRNA that is assayed? What type of assay? This should be improved.

There is some significant repetition in that should be edited.

Generally, the writing is good, but some English specific editing would improve the clarity and flow of this manuscript.

Author Response

Dear Reviewer 4.

Round 2

Reviewer 2 Report

The authors have sincerely revised the manuscript according to the reviewer's comments.

Author Response

Dear Reviewer 2,

Thank you in advance.

Reviewer 3 Report

The authors improved the article, corrected many of the mistakes made. However, I would have some more observations.

The definition of muscular dystrophies is not correct. Most muscular dystrophies (DM) appear as a consequence of a genetic mutation that affects a muscle protein, which is no longer synthesized or is synthesized incorrectly, with a modified molecule. There are forms of DM that manifest in adulthood. So the authors' definition is somehow forced, it is not mainly about myogenesis, metabolism or ion channels. Sure, the affected proteins can be part of ion channels or they can be enzymes, but most of them are structural protein of the muscle fiber, the myogenesis process takes place, only that the resulting muscle fibers no longer have a normal physiology (I won't go into details). Not all forms of DM have “a catastrophic progression of muscle wasting and massive loss of muscle regeneration, producing fragile fibers due to impaired myogenesis, leading to premature death”.  The form described here is only for dystrophinopathies, respectively Duchenne/Becker.

Articles that do not refer to muscular dystrophies related to Pax7 should not be included in the analysis because they do not refer to the topic. There are many articles treating the role of Pax7 in myogenesis, but which do not refer to the topic chosen by the authors.

Many of the English language mistakes have been corrected.

Author Response

Dear Reviewer 3,

I thank you in advance.
